behaviour, cognition, evolution

what–where–when memory, ageing, semantic-like memory, time-place learning, cephalopod cognition, comparative cognition

**Author for correspondence:**
Alexandra K. Schnell
e-mail: alex.k.schnell@gmail.com

# Episodic-like memory is preserved with age in cuttlefish

Alexandra K. Schnell[1,2,3], Nicola S. Clayton[2], Roger T. Hanlon[3] and Christelle Jozet-Alves[1]

[1]Normandie Univ., UNICAEN, Univ Rennes, CNRS, UMR EthoS 6552, Caen, France
[2]Department of Psychology, University of Cambridge, Cambridge, UK
[3]Marine Biological Laboratory, Woods Hole, MA 02543, USA

 AKS, 0000-0001-9223-0724; NSC, 0000-0003-1835-423X; RTH, 0000-0003-0004-5674; CJ-A, 0000-0002-9372-2306

Episodic memory, remembering past experiences based on unique what–where–when components, declines during ageing in humans, as does *episodic-like* memory in non-human mammals. By contrast, semantic memory, remembering learnt knowledge without recalling unique what–where–when features, remains relatively intact with advancing age. The age-related decline in episodic memory likely stems from the deteriorating function of the hippocampus in the brain. Whether episodic memory can deteriorate with age in species that lack a hippocampus is unknown. Cuttlefish are molluscs that lack a hippocampus. We test both semantic-like and episodic-like memory in sub-adults and aged-adults nearing senescence ($n = 6$ per cohort). In the semantic-like memory task, cuttlefish had to learn that the location of a food resource was dependent on the time of day. Performance, measured as proportion of correct trials, was comparable across age groups. In the episodic-like memory task, cuttlefish had to solve a foraging task by retrieving what–where–when information about a past event with unique spatio-temporal features. In this task, performance was comparable across age groups; however, aged-adults reached the success criterion (8/10 correct choices in consecutive trials) significantly faster than sub-adults. Contrary to other animals, episodic-like memory is preserved in aged cuttlefish, suggesting that memory deterioration is delayed in this species.

## 1. Introduction

Episodic memory is the ability to remember unique past events [1]. This type of memory receives and stores information about the context in which an event occurred including *what* happened, *where* and *when*. In humans, episodic memory develops around the age of 4 [2–6] and declines with advancing age [7–11]. This type of memory differs from another form of recollection, namely semantic memory, which is the ability to recall general knowledge acquired through learning without retrieving unique spatio-temporal features about the learning context. Unlike episodic memory, semantic memory remains relatively intact with advancing age [12–17].

Episodic memory was once thought to be uniquely human because verbal reports from humans suggest that retrieving personally experienced events are accompanied by the conscious experience of recollection, the so-called autonoetic consciousness [18]. The idea that episodic recall requires autonoetic consciousness represents an intractable barrier for demonstrating episodic memory in animals, since conscious experience cannot be evaluated in non-verbal subjects in the absence of agreed behavioural markers of non-linguistic consciousness [19]. This dilemma was partially resolved by focusing on features of episodic recall that can be demonstrated behaviourally without the need for language [20–22]. The behavioural criteria developed by Clayton & Dickinson [20] were designed to test whether food-caching jays could solve a task by remembering the what–where–when components of a specific past caching event. These behavioural

criteria were able to demonstrate that jays can simultaneously retrieve spatio-temporal features of a specific past event. However, Clayton & Dickinson [20] also acknowledged that, in the absence of any agreed behavioural markers of consciousness, it is not possible to evaluate whether memory retrieval in jays is also associated with conscious experience of remembering. As such, Clayton & Dickinson [20] coined the term *episodic-like* memory to refer to the ability to remember the what–where–when of unique past episodes, in the absence of any demonstration that this memory was accompanied by the conscious awareness of pastness (chronesthesia) and authorship (of being the owner of the memory, i.e. autonoesis).

In a similar vein, the term semantic memory has been confined to humans because it implies that remembering learnt knowledge is related to meaning or reference in language. For this reason, neuroscientists often refer to the recollection of learnt knowledge in animals as reference memory. Reference memory, however, is not normally couched in terms of what–where–when memory and is typically defined as a type of memory for information that is held constant over time, an attribute that conflicts with semantic memory because semantic knowledge can be updated with new information. Consequently, here we will refer to semantic memory in animals as *semantic-like* memory. Note that studies on humans show that episodic memory is embedded within semantic memory, i.e. that semantic memory serves as a semantic scaffold and context to support the experiential aspects of episodic recall [18,23].

Behavioural criteria for episodic-like memory have now been used to show that closely related corvids such as magpies [24] as well as distantly related taxa from non-human apes [25] (but see [26]) and rodents [27,28] to zebrafish [29] and cuttlefish [30] also possess the ability to recall the what–when-where of unique past episodes. Despite this growing body of evidence, the commonality between episodic-like memory in animals and episodic memory in humans has been a topic of much debate [31–33] (but see [34,35]). How similar might episodic memory in humans be to that in a rat? In humans, the coding of episodic information can be specified in time rather than space. We might mentally recall a memory of a party, but the order of the memory is not itemized by different locations. A similar phenomenon has been discovered in rats, based on their sophisticated discrimination of different odours. Specifically, rats remembered many different episodes by the order in which they occurred rather than by the different locations in which they smelt the odours [36]. Rats have also been found to replay a stream of multiple episodic memories to remember the order in which the events occurred [37]. Among avian species, scrub-jays possess similar features to episodic memory in humans because their episodic-like memory is embedded within semantic-like memory and deployed flexibly, both hallmarks of this type of memory in humans. For example, scrub-jays were able to integrate episodic-like information about unique past caching episodes with semantic-like information about the perishability rates of the food types they had cached. Moreover, when new information was acquired many hours after the memory had been encoded, they were able to update that information accordingly and use it to inform and change their decisions about which caches to retrieve at a later date [22,38,39].

While increasing evidence suggests that some animals show hallmarks of an episodic memory system, less is known about its development through the lifespan of animals.

Specifically, there is limited evidence of the effects of physiological ageing on episodic-like memory. Using an integrated what–where–when paradigm, a single study on mice has shown age-related decline in episodic-like memory [40]. Other studies on monkeys and rodents show that both spatial (what–where) [41–43] and temporal features of memory (what–when) [44] appear to decline as a result of age-related changes in the brain. These memory impairments have primarily been attributed to alterations in medial–temporal lobe regions and the hippocampus, a brain structure that plays a vital role in learning and memory. Structural [45], neurochemical [46] and functional alterations are indeed associated with impairments in hippocampus-dependent cognitive processing. Age-related changes to the hippocampus are remarkably similar across taxa with mammalian brain architecture [47]. Thus, it might be assumed that animals with similar brain structures that possess episodic-like memory are susceptible to the same effects of physiological ageing. But what about animals capable of episodic-like memory but with significantly different brain structures? Can episodic-like memory deteriorate with age in taxa that lack a hippocampus?

Cuttlefish exhibit dramatic differences in both brain structure and brain organization compared to vertebrates [48–51]. They lack a hippocampus and instead possess a vertical lobe. Yet the vertical lobe presents similarities in connectivity and functionality with the hippocampus formation in vertebrates [52–54], it is the epicentre for both learning and memory. Common cuttlefish, *Sepia officinalis,* also possess episodic-like memory. They can optimize their foraging behaviour by remembering unique foraging events based on what they have eaten, where they have eaten it and how long ago [30]. However, little is known about whether advancing age impairs memory in common cuttlefish—a question that is highly amenable to investigation because they have short lifespans (approx. 2 years of age) [55]. Currently, two studies have shown that learning during an associative learning task is preserved in aged cuttlefish (22 months old), whereas long-term memory retention is impaired [56,57]. Specifically, aged cuttlefish nearing senescence were able to learn at the same rate as younger cuttlefish but, unlike the younger subjects, aged cuttlefish could not successfully complete the task following a 24 h delay.

To investigate whether the episodic-like memory system in cuttlefish is vulnerable to age-related impairments, we tested two cohorts, sub-adults (10–12 months) and aged-adults nearing senescence (22–24 months), and compared performance in two distinct memory tasks: a semantic-like memory task and an episodic-like memory task. The difference between these tasks centres on the uniqueness of the memory recollection. In the semantic-like memory task subjects were required to recall learnt information that was not unique to a specific time or place but was fixed across training and test days. By contrast, in the episodic-like memory task subjects were required to recall a specific memory that was unique to each test day. These tasks were chosen to investigate whether, like humans, only episodic information deteriorates with age while semantic information stays relatively intact.

## 2. Methods

### (a) Subjects

The experiments took place at the Marine Biological Laboratory, Woods Hole, USA (41°31′ N, 70°39′ W). Twenty-four cuttlefish

were used in this study. Subjects had not participated in any cognitive experiments prior to this study. Cuttlefish were reared from eggs in the Marine Resources Centre at the Marine Biological Laboratory (see electronic supplementary material).

We used two cohorts of cuttlefish in this study including sub-adults (10–12 months of age) and aged-adults (22–24 months of age). Throughout the experiments, subjects were housed individually in fibreglass tanks, which were supplied with a constant flow of filtered natural seawater (approx. 10 l min$^{-1}$) maintained under natural daylight conditions and at a temperature of 16–17°C (see electronic supplementary material). Outside of the training and testing period, cuttlefish were fed a mixed diet of food items including pieces of thawed penaeid prawn meat, live grass shrimp, *Palaemonetes paludosus*, live gammarid crustaceans, *Platorchestia platensis*, and juvenile Asian shore crabs, *Hemigrapsus sanguineus* (see [58] for specific food quantities per subject size). Feeding took place three times daily at roughly 9.00 h, 12.00 h and 16.30 h.

## (b) Procedure

To ensure the cuttlefish were mildly hungry and thus motivated to participate in the food-rewarded experiments, the 9.00 h and 12.00 h feeding sessions were omitted during experimental periods. The amount of food they obtained during training and testing replaced the food acquired during these non-experimental feeding times. If a subject did not participate in the trials the subject was offered extra food during the 16.30 h feed (see the electronic supplementary material).

## (c) Pre-training: associating a visual cue with an edible reward

Prior to the experiments, we trained cuttlefish to approach a specific location in their tank, marked with a visual cue. Visual cues consisted of a black and white PVC square (25 × 25 mm; *l* × *h*) attached to an extendable rod that could be mounted onto the side of their tank. Subjects were required to approach the visual cue, at a distance of least 10 cm, within a 60 s period following presentation. Subjects were trained until they reached a success criterion of at least eight correct choices in 10 consecutive trials (see electronic supplementary material).

## (d) Semantic-like memory

To test for age-related decline associated with semantic-like memory, we trained sub-adult cuttlefish at the age of 12 months ($n = 6$) and aged-adult cuttlefish nearing senescence ($n = 6$) at the age of 24 months to complete a what–where–when task that could be solved by applying learnt semantic-like information. In this task, the cuttlefish were required to visit three different locations in their tank over a 6 h period, with only one location rewarded with a food item every 3 h. Thus, a total of three feeding sessions were offered, a 'breakfast', 'lunch' and 'dinner' feed. To receive each feed, cuttlefish had to learn that the location of the rewarded food resource was dependent on the time of day. Each feeding location was separated from each other by 30–40 cm (i.e. 30 cm for sub-adults and 40 cm for age-adults) but was marked with identical visual cues. The subjects were trained across multiple days over three sessions per day at fixed time-points (e.g. 9.00 h, 12.00 h, 15.00 h) so that they learnt to visit or avoid specific locations at certain times. Across the cuttlefish, the order in which the three locations were rewarded was pseudo-randomly assigned. The relative position and the order of rewards across the different locations differed for each cuttlefish (note that each cuttlefish was tested within its own individual tank). Each subject was assigned a starting point in their tank that offered visual access to all three chosen locations that were

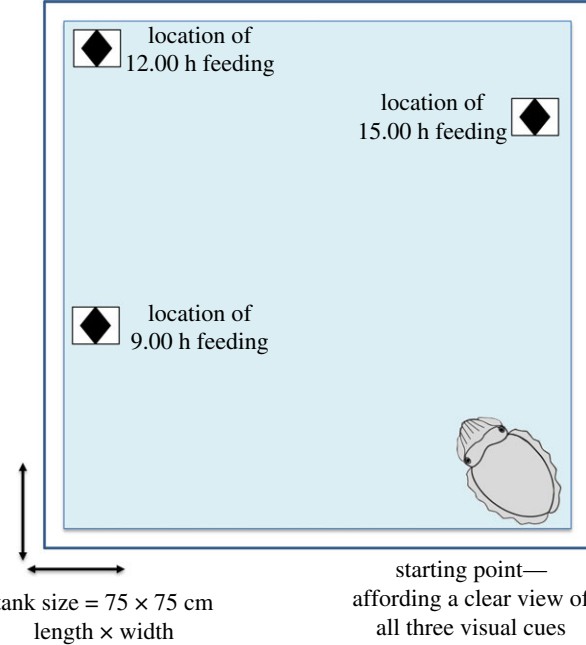

**Figure 1.** Representation of the test phase of the semantic-like memory experiment, depicting sample starting point for the cuttlefish and sample visual cue configuration (not drawn to scale). Note that in the test phase of this experiment the visual cue configuration was fixed across all training and test days. (Online version in colour.)

to be marked with the visual cues (figure 1). At the start of each trial, subjects were gently ushered to their starting positions using a soft mesh net.

On the first day and second day of the experiment, each subject experienced sequential cue presentation. For the 'breakfast' feed, we presented a single visual cue at the first location and allowed the cuttlefish to approach the location to receive a food reward (i.e. either a portion of prawn meat or a live grass shrimp). After 5 min, we removed the first visual cue and returned 3 h later for the 'lunch' feed to present the second visual cue at the second location. This process was repeated for the 'dinner' feed with the third visual cue at the third location.

On the third day of the experiment, we presented cuttlefish with all three visual cues simultaneously placed at their assigned locations. Only the location designated as the 'breakfast' feed for that cuttlefish was rewarded during this first presentation. The other two locations marked with visual cues did not lead to a reward. Following a 5 min period, all visual cues were removed, and the process was repeated 3 h and 6 h later. Each cuttlefish had a total of 5 min to decide which visual cue to approach. We determined that the cuttlefish had made a decision when it approached a visual cue at a distance of at least 10 cm with the centre of their head aligned with the visual cue (see electronic supplementary material, figure S1). Each cuttlefish experienced simultaneous cue presentation for 15 days within a three-week period. During each trial, we recorded all visits made by the cuttlefish, noting both the location and time of day.

To determine whether the cuttlefish used the time of day or the sequence in which they visited each location to obtain the edible reward, we carried out three consecutive test days, which commenced the day after training had finished. In this test, we either omitted the first feeding session or the first two feeding sessions, and presented the visual cues at what had been the third hour or the sixth hour of the normal sessions, the assigned time for the 'lunch' and the 'dinner' feed,

respectively. This type of test, whereby the first session (i.e. the 'breakfast' feed) is omitted, allowed us to determine whether the subjects used a daily ordinal timing strategy or whether they were able to base their decisions on the time of day [59]. If the cuttlefish approached the first visual cue regardless of the time of day this would signify that the cuttlefish were using the order in which the locations were rewarded and not the time of day. By contrast, if cuttlefish approached the third visual cue that rewarded the dinner feed this would indicate that the cuttlefish were basing their decisions on the time of day and thus capable of associating a specific location with a specific time (i.e. time–place learning) [59].

### (e) Episodic-like memory

To test for age-related decline associated with episodic-like memory, we trained sub-adult cuttlefish at the age of 10 months ($n = 6$) and aged-adult cuttlefish nearing senescence ($n = 6$) at the age of 22 months to complete a what–where–when task that could only be solved by retrieving *what–where–when* information about a past event with unique spatio-temporal features.

#### (i) Prey preferences

We conducted tests to determine the prey preferences of each individual subject. Cuttlefish were simultaneously presented with two different prey types that were equal in size: live grass shrimp and a portion of prawn meat. Each prey item was fixed to a clear plastic dowel stick (1 mm diameter), thus restraining but not immobilizing the live prey. Both dowel sticks were then presented in pseudo-random locations at equal distances to the cuttlefish. Each cuttlefish received 20 food preference trials over 10 consecutive days. The item that they approached initially was given to the cuttlefish and rated as the preferred prey item and the alternative prey item was immediately removed.

#### (ii) Training phase: replenishing rates of different prey types

Cuttlefish were trained to learn that the different prey types were available for consumption at specific locations and after specific delays (i.e. 1 h or 3 h delay). Specifically, less preferred prey types were replenished after 1 h, but preferred prey types were only replenished after 3 h. In this phase, cuttlefish were offered two feeding sessions. In the initial feeding session, cuttlefish were given a choice between both prey types, whereby two identical visual cues were presented in two distinctive locations. When a cuttlefish randomly approached one of the cues, both prey types were simultaneously presented in front of their respective cue (note each prey type is location specific). The cuttlefish was then allowed to capture one of the two prey types.

Subjects were then offered a second feeding session, either 1 h later or 3 h later. Each cuttlefish received a single pseudo-randomized delay trial each day, either a 1 hr delay or a 3 h delay trial. In the 1 h delay trials, cuttlefish were trained to learn that their preferred prey had not been replenished and thus was not available, only the less preferred prey was available for consumption. In the 3 h delay trials, cuttlefish were trained to learn that both prey types had been replenished and were available for consumption. In the delayed trials, prey items were presented only after the cuttlefish had approached a cue and were only rewarded if the subject had chosen correctly. This ensured that the subjects were not using the sight or smell of the prey to guide their decision-making because the prey was only placed in the water once the cuttlefish had made a choice. In the 1 h delay trial, a choice was considered correct if the cuttlefish avoided the visual cue marking the location associated with their preferred prey and approached the location associated with their less preferred prey. In the 3 h delay trial, a choice was considered correct if the cuttlefish approach the visual cue marking

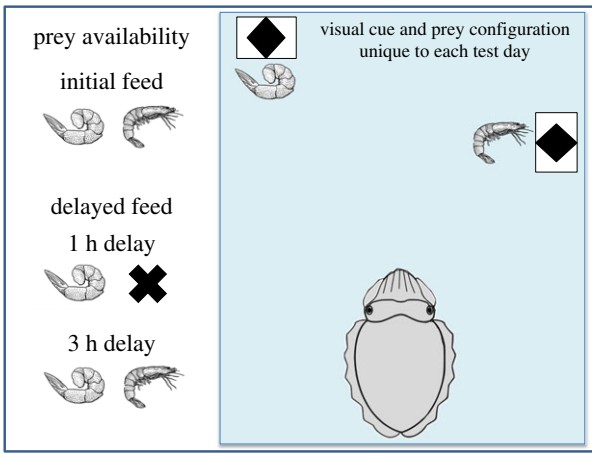

**Figure 2.** Representation of the test phase of the episodic-like memory experiment, depicting the availability of each prey type and their replenishing rates after a 1 h and a 3 h delay (not drawn to scale). Note that in the test phase of this experiment the visual cue and prey configuration were unique to each test day. (Online version in colour.)

the location associated with their preferred prey. Subjects were trained until they reached a success criterion of at least eight correct choices in 10 consecutive trials.

#### (iii) Testing phase: recollecting spatio-temporal features of a unique foraging event

To test for episodic-like memory, cuttlefish were presented with the same process outlined in the training phase, except that the visual cues were placed in unique locations on each test day. Specifically, each day, a location marked by a visual cue was pseudo-randomly assigned to the prey items. This daily assignment was maintained for both the initial and the delayed feeding session. Consequently, for the subjects to access the correct prey type in the delayed feeding trial cuttlefish were required to retrieve unique spatio-temporal information based on what they had initially eaten, where they had eaten it and how much time had lapsed since their initial feeding (when) (figure 2).

### (f) Analysis

All data were analysed with non-parametric tests and computed using R software (v. 3.5.1). For the semantic-like memory task, we used a non-parametric analysis of variance to compare the number of correct trials between the different age groups. In the training phase, we analysed trials through time (i.e. blocks of 3 days) as well as the number of sessions necessary to reach 8 out of 10 consecutive correct choices. In the test phase, we analysed trials per day across three consecutive test days. We also used a binomial test to determine whether cuttlefish were basing their decisions on ordinal timing or on the time of day. Proportions for the binomial tests were tested against a value: 0.33 as subjects had to choose to approach one of three locations.

For the episodic-like memory task, we used binomial tests to determine whether cuttlefish had preferences for different prey types. We then used an exact permutation test for independent samples to determine whether prey preferences differed between the different age groups. We also used exact permutation tests for independent samples to compare the number of days to reach the success criterion (i.e. at least eight correct choices in 10 consecutive trials) between the different age groups in both the training phase and the test phase. We then used a non-parametric analysis of variance to compare the proportion of correct trials in the test phase between the different age groups with both age and delay type (1 h or 3 h) as independent variables.

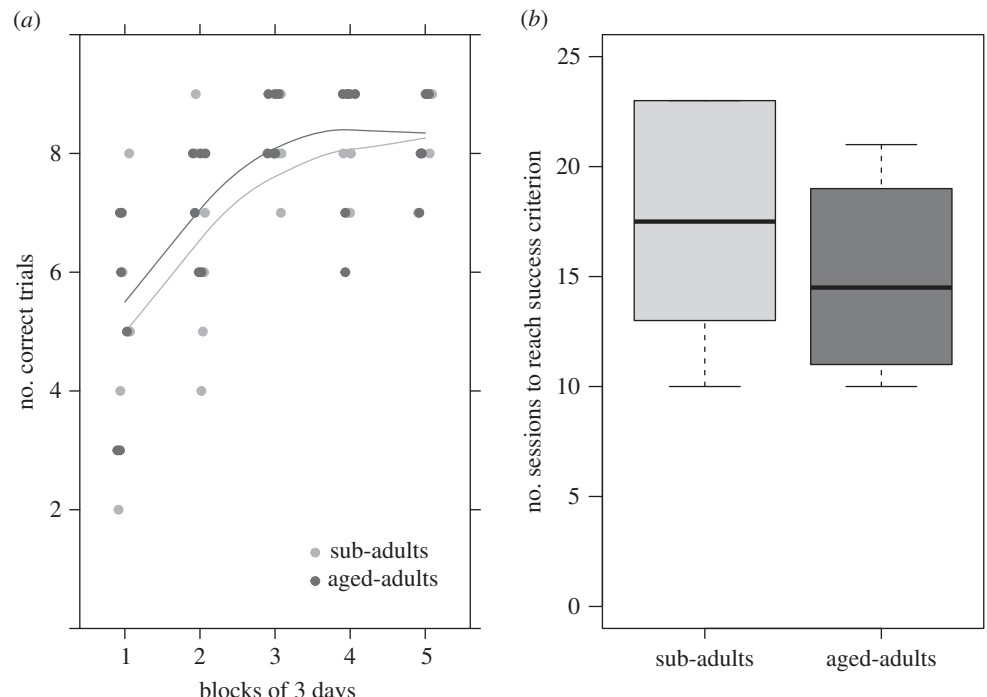

**Figure 3.** Performance during the training phase in the semantic-like memory task in sub-adult ($n = 6$) and adult ($n = 6$) cuttlefish. (*a*) The mean number of correct trials across blocks of 3 days) and (*b*) the mean number of sessions necessary to reach the success criterion (8 out of 10 correct consecutive choices).

## 3. Results

### (a) Semantic-like memory

In the training phase, the mean number of correct trials increased across time (i.e. blocks of 3 days; non-parametric analysis of variance: $p < 0.001$). There was no significant difference between age groups ($p = 0.593$) and no significant interaction between time and age ($p = 0.754$) (figure 3*a*). The mean number of sessions necessary to reach 8 out of 10 correct consecutive choices did not differ significantly between age groups (exact permutation test for independent samples: $Z = -0.89$; $p = 0.37$; figure 3*b*). In the test phase, the majority of the cuttlefish successfully passed the test across three consecutive days (binomial test: 5/6 cuttlefish with 1/3 probability of success; $p = 0.001$; CI = 0.54–1.00; 6/6 cuttlefish with 1/3 probability of success; $p = 0.018$; CI = 0.36–0.99; figure 4). There was no effect of age (non-parametric analysis of variance: $p = 1$) or day ($p = 0.817$) and no interaction between age and day ($p = 0.558$).

### (b) Episodic-like memory

All cuttlefish except for one individual showed a strong preference for live grass shrimp over portions of prawn meat (at least 15 live shrimp out of 20 choices). This preference was comparable between age groups (exact permutation test for independent samples: $Z = -0.567$; $p = 0.584$; electronic supplementary material, figure S2).

In the training phase, we found no significant difference in number of days to reach the success criterion between age groups (exact permutation test for independent samples: $Z = 0.818$; $p = 0.413$; figure 5*a*). By contrast, in the test phase, adults reached the success criterion significantly faster than sub-adults (exact permutation test for independent samples: $Z = -2.307$; $p = 0.021$; figure 5*a*). However, performance— measured as proportion of correct trials during the test phase—was comparable between age groups (non-parametric

analysis of variance: $p = 0.211$) and across the different delays ($p = 0.179$) and there was no significant interaction between age and delay ($p = 0.492$) (figure 5*b*).

## 4. Discussion

Our results suggest that episodic-like memory in cuttlefish does not decline with age, unlike that observed in humans. Both sub-adult and aged-adult cuttlefish successfully completed both the semantic-like and episodic-like memory tasks. In the semantic-like memory task, cuttlefish were able to learn the location of a rewarded food resource in response to the time of day. The number of correct choices and the number of sessions to successfully complete the task did not differ across the age groups. In the episodic-like memory task, cuttlefish were able to remember unique what–where–when components of previous foraging events to guide their foraging decisions. Specifically, they were able to remember what they had eaten for a unique breakfast event, where they had eaten it, and how long ago. Subjects used this spatio-temporal information to guide decisions about whether to search for highly preferable prey (live shrimp) that was replenished after long delays (3 h) or less preferred prey (prawn meat) that was replenished after short delays (1 h). The number of days taken to learn the replenishing rates of the different prey types did not differ across the different age groups. Likewise, the number of correct choices in the test phase did not differ between sub-adult and aged-adult cuttlefish. However, the number of sessions taken to successfully complete the episodic-like memory task in the test phase differed across the age groups. Aged-adult cuttlefish were able to successfully complete the test phase faster than sub-adult cuttlefish.

In humans, memory deterioration is most often associated with normal ageing and episodic memory is typically the earliest and most strikingly affected by ageing [10,12,60,61]. Episodic-like memory in non-human primates and rodents

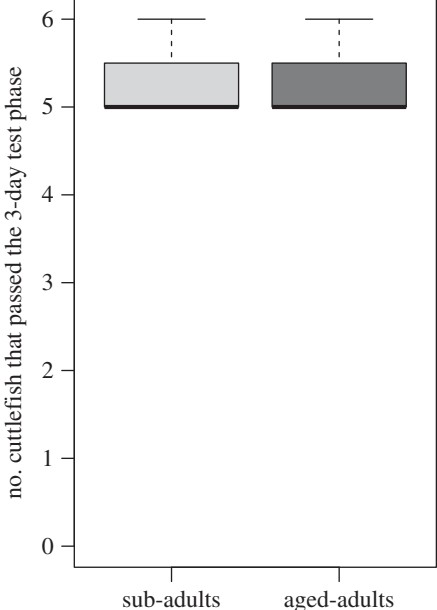

**Figure 4.** Performance during the test phase in the semantic-like memory task in sub-adult ($n = 6$) and aged-adult ($n = 6$) cuttlefish. The graph represents the number of cuttlefish that chose the correct location across the three test days.

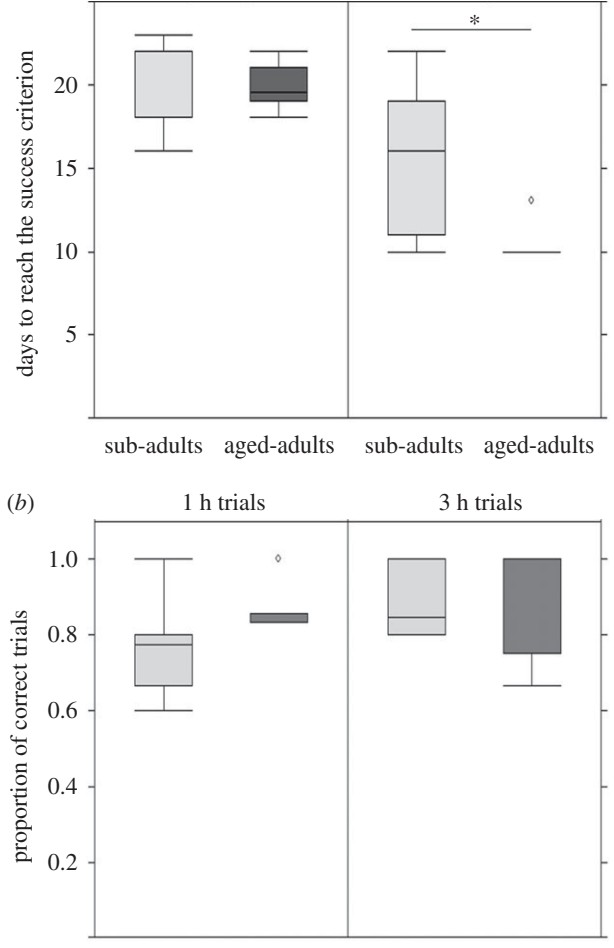

**Figure 5.** Performance during the episodic-like memory task in sub-adult ($n = 6$) and aged-adult ($n = 6$) cuttlefish. (*a*) The mean number of days ± s.e. number to reach the success criterion (8 out of 10 correct consecutive choices) in the training and test phase of the experiment; and (*b*) the mean proportion of correct trials ± s.e. during the delayed feeding sessions in the test phase.

is also vulnerable to the effects of physiological ageing [39–41,43,62,63]. While evidence in animal models is limited, the present work provides the first evidence of an animal that appears to be resistant to age-related deterioration at least within the context of episodic-like memory.

These findings are surprising given that, similar to mammalian species, the high metabolic rate of cephalopods correlates with a high rate of radical production [64], which can lead to oxidative stress and contribute to physiological ageing. Furthermore, compared to mammalian species, common cuttlefish and Atlantic brief squid, *Lolliguncula brevis*, show significantly lower levels of enzymatic antioxidant defence [64], a process that can counter oxidative damage. This relatively low antioxidative status found in these cephalopod species has been linked to their short life expectancy [64].

Does this mean that cuttlefish preserve spatio-temporal features of memory despite showing expedited physiological signs of ageing? Comparable performance between sub-adults and aged-adults in the semantic-like memory task suggest that spatial features of learning and memory are preserved in aged cuttlefish. In a similar vein, comparable performance between the different age groups in the episodic-like memory task suggest that unique and integrated spatial and temporal components of memory are also preserved in aged cuttlefish. These results are in line with previous research that shows that learning performance in an associative learning task with negative reinforcement was comparable between young and aged cuttlefish [57]. While our results indicate that cuttlefish show comparable performance in both memory tasks regardless of age, notice that the tasks include a different number of delay conditions. Specifically, the semantic-like memory task involves three delay conditions (i.e. 9.00 h, 12.00 h, 15.00 h), whereas the episodic-like memory task involves only two delay conditions (i.e. 1 × breakfast feeding and 1 × delay feeding). The difference in the number of delay conditions across both tasks means that we were unable to equate the complexity of the tasks, making it difficult to draw

comparisons of performance. Future research could increase comparability by designing tasks that involve an equal number of delay conditions. It is also important to note that we used a sample size of six subjects per cohort for each experiment. Increasing sample sizes in future studies could improve the evidential value of our current findings.

Note that cuttlefish are not completely resistant to age-related memory decline. Behavioural and neuro-histological investigations in cuttlefish have revealed significant changes in memory retention and obvious signs of age-related brain degeneration [56,57]. For example, aged cuttlefish show significantly poorer scores in memory retention (after a 24 h delay) suggesting that their long-term memory processes deteriorate with age [56,57]. Aged cuttlefish also show numerous signs of degeneration in the superior frontal, sub-vertical and pre-commissural lobes—brain structures that are implicated in the input and output mechanisms of memory storage [56]. By contrast, the vertical lobe, involved in learning and memory processing [49,53,54], is highly resistant to age-related degeneration [56]. However, this preservation is mainly confined to inside the neuropile of the vertical lobe. There are numerous fibres between the vertical lobe and sub-vertical lobe that degenerate with age [56]. As a result of this

degeneration, the vertical lobe might not be able to receive or send properly processed pre-treated information. This type of degeneration only appears in the final days of an individual's life, once the cuttlefish has completely stopped eating (i.e. 1–2 days before perishing; C.J.-A. 2010, personal observation). In the present study, aged cuttlefish died from senescence $25.67 \pm 3.82$ (mean number of days $\pm$ s.e.) days after the completion of the tasks. While we cannot exclude that a sharp decline of spatio-temporal memory abilities occurs in the last days of life in cuttlefish, it is likely that cognitive ageing in this species does not follow a gradual time course as in other studied taxa. For instance, in mammals, episodic memory declines steadily across adulthood [65]. The preservation of episodic-like memory in aged cuttlefish might be attributed to the delayed levels of physiological degeneration that only appears to occur when death is imminent.

Why would cephalopods preserve this type of complex memory system (i.e. show no evidence of age-induced memory deterioration)? Perhaps the answer lies in their need to overcome evolutionary pressures. In a recent review, Amodio and colleagues [66] suggested that a combination of increased predation, enhanced foraging challenges, and intense mating competition [67,68] triggered the emergence of large brains and behavioural complexity in cephalopods. Such pressures, particularly mating competition, which occurs later in their life cycle, might have also played a role in their apparent resistance to age-related decline with regard to complex learning and memory processes. Consider that most cephalopods species have a single reproductive period in which individuals mate across several temporally finite instances [68,69]. Limited breeding periods combined with highly skewed operational sex ratios experienced by various species results in fierce mating competition [70–72]. This period of intense competition involves aged-adults and is shortly followed by senescence [69]. It has been suggested that mating competition in cephalopods gave rise to some aspects of complex cognition as competing individuals must outwit group members to monopolize more resources (i.e. members of the opposite sex) [73]. Resistance to age-related decline to preserve complex learning and memory may have been positively selected for because it enables individuals to recall specific spatio-temporal features of past mating events, especially given the relatively short lifespan of these animals. Recollecting this knowledge could help individuals optimize their mating behaviour during these finite breeding periods just prior to entering the short stage of senescence. Future research might focus on episodic-like memory in cuttlefish within a mating context and determine whether older subjects preserve episodic-like information to guide mating decisions.

In conclusion, these results suggest that the episodic-like memory system in cuttlefish differs from episodic-like memory in other non-human species, at least in terms of its development across the lifespan of the animal. Whether this difference is a result of the different neuroanatomy possessed by cuttlefish requires further attention. Moreover, future research could highlight whether the development of episodic-like memory is delayed in cuttlefish or whether individuals possess this type of memory shortly after hatching. Overall, these findings highlight the common cuttlefish as an interesting model for investigating resistance to age-related decline in the episodic-like memory system. While finer-grained analyses of the effects of physiological ageing on memory formation, processing, and retention in cuttlefish are needed, these initial results suggest that cuttlefish are valuable models for investigating the natural mechanisms that protect complex memory from the effects of ageing.

Ethics. Ethical approval was not required for the experiments as there are currently no ethical regulations in place for research on cephalopods in the USA. Nevertheless, this research followed the guidance given by Directive 2010/63/EU, the guidelines for the care and welfare of cephalopods in research [74] and complied with the Animal Research: Reporting In Vivo Experiments (ARRIVE) guidelines. Moreover, the study was designed with the intention of minimizing potential stress to the animals and noxious stimuli were not used in any of the experiments.

Data accessibility. All data needed to evaluate conclusions in the manuscript are present as raw data in the electronic supplementary material [75].

Authors' Contributions. A.K.S.: conceptualization, data curation, formal analysis, investigation, methodology, project administration, validation, writing-original draft; N.S.C.: conceptualization, methodology, supervision, validation, writing-review and editing; R.T.H.: methodology, resources, supervision, writing-review and editing; C.J.-A.: conceptualization, formal analysis, methodology, supervision, validation, writing-review and editing.

All authors gave final approval for publication and agreed to be held accountable for the work performed therein.

Competing interests. We have no competing interests, and all authors declare that the research was conducted in the absence of any commercial or financial relationships that could be construed as a potential conflict of interest.

Funding. This work was supported by multiple funding bodies during the conception, development, data collection and writing of this study. A.K.S. was supported by a post-doctoral study grant from the Fyssen Foundation, an Endeavour Research Fellowship funded by the Australian Government, the Grass Fellowship Program funded by the Grass Foundation and by a Newton International Fellowship funded by the Royal Society. N.S.C. was supported by a European Research Council under the European Union's Seventh Framework Programme (FP7/2007–2013)/ERC grant agreement no. 3399933. R.T.H. was supported partially by the Sholley Foundation.

Acknowledgements. Many thanks to the funding bodies that supported the authors. We thank the technical staff and the intern staff at the Marine Biological Laboratory in Woods Hole, MA USA for support with aquaria maintenance and animal husbandry. We offer thanks to the members of the Comparative Cognition Lab for helpful discussion about the experimental design.

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
