## [Peer Review File · Proceedings of the Royal Society B: Biological Sciences]

Review History

RSPB-2021-1052.R0 (Original submission)

Review form: Reviewer 1

Recommendation

Major revision is needed (please make suggestions in comments)

Scientific importance: Is the manuscript an original and important contribution to its field?

Excellent

General interest: Is the paper of sufficient general interest?

Excellent

Quality of the paper: Is the overall quality of the paper suitable?

Acceptable

Is the length of the paper justified?

Yes

Should the paper be seen by a specialist statistical reviewer?

No

Do you have any concerns about statistical analyses in this paper? If so, please specify them explicitly in your report.

No

It is a condition of publication that authors make their supporting data, code and materials available - either as supplementary material or hosted in an external repository. Please rate, if applicable, the supporting data on the following criteria.

Is it accessible?

Yes

Is it clear?

No

Is it adequate?

No

Do you have any ethical concerns with this paper?

Yes

Comments to the Author

I read the manuscript “Episodic-like memory is preserved with age in cuttlefish” and found it an interesting study exploring episodic memory in the cephalopod mollusc *Sepia officinalis*.

First of all, I have to say that despite the study does not require ethical approval because experiments on live animals have been carried out in the USA, there are important details about care, maintenance and use of live animals for research purposes that should be reported. This is mandatory according to ARRIVE Guidelines – that this Journal and the Royal Society that states “Authors are expected to comply with the 'Animal Research: Reporting In Vivo Experiments' (ARRIVE) guidelines. These have been developed by NC3Rs to improve standards of reporting to ensure that the data from animal experiments can be fully scrutinised and utilised. Relevant information should be included in the appropriate section of the article, as outlined in the ARRIVE guidelines” (see <https://royalsociety.org/journals/ethics-policies/research-ethics/>).

I strongly recommend the Authors to revise their manuscript attentively, in order to report all the necessary details about this study as required by ARRIVE Checklist and the best practice of in vivo studies reporting (see also PREPARE: Smith, A.J., Clutton, R.E., Lilley, E., Hansen, K.E.A., and Brattelid, T. (2018). PREPARE: guidelines for planning animal research and testing. *Laboratory Animals* 52(2), 135-141. doi: 10.1177/0023677217724823.). I recommend to provide details (to mention some) about: i. the individual/group housing of the animals between experimental trials; ii. body size of the animals (in addition to age group); iii. size and shape and all required details about the holding and experimental tanks; iv. monitoring of seawater quality, v. number of subjects excluded (if any).

I would like also to remind that following the “Guidelines for the treatment of animals in behavioural research and teaching” studies carried out using captive animals “may mean obtaining them from the wild and necessarily involve confinement” require attention in assuring best practice and compliance with the principles of keeping animals in an optimal welfare status, including manipulation/handling.

This reviewer is fully aware of the long, well established, consolidated practice in housing live cephalopods at the MBL, but I reiterate that the overall quality of the scientific work using these species will enormous improve if transparency about animal care and maintenance and details about experimental settings will be provided, despite not required by legislation in non-EU countries.

The study is an interesting one exploring time-age capacity for episodic and semantic memories in the cuttlefish. As reported by Authors, known age-related decline in episodic memory “is thought to stem from deteriorating function of the hippocampus in the brain. Whether episodic memory can deteriorate with age in species that lack a hippocampus”. Testing this research

question in cuttlefish offers interesting advantages.

Authors tested semantic- and episodic-like memories in sub-adults and aged cuttlefish. The semantic-like memory task was designed to let the animal to learn that the location of a food resource in relation with the time of the day. In the episodic-like memory task, the paradigm designed required the solution of a foraging task (what-where-when information about a past event with unique spatiotemporal features). Performance of the semantic-like memory task resulted comparable across age groups. However, with the episodic memory Authors found that aged-adults reached the success criterion faster than sub-adults. Authors conclude that contrary to what known in other species, episodic-like memory is preserved in aged cuttlefish, “suggesting that memory deterioration is delayed in this species”.

Independently from what anticipated in the previous paragraphs, I am in the unfortunate situation of not being able to recommend the acceptance of the manuscript in its current form. I am going to provide the following comments for Authors to consider in their substantial revision.

1. The ‘statement’ (line 28) “Cuttlefish are molluscs that lack a hippocampus” is not accurate. Despite the classic knowledge by Young and colleagues, that I recommend to refer to, I suggest Authors to reconsider their statement and refer to:

Shomrat, T., Graindorge, N., Bellanger, C., Fiorito, G., Loewenstein, Y., and Hochner, B. (2011). Alternative Sites of Synaptic Plasticity in Two Homologous "Fan-out Fan-in" Learning and Memory Networks. *Current Biology* 21, 1773-1782.

Shigeno, S., Andrews, P.L.R., Ponte, G., and Fiorito, G. (2018). Cephalopod Brains: An Overview of Current Knowledge to Facilitate Comparison With Vertebrates. *Front. Physiol.* 9, 952. doi: 10.3389/fphys.2018.00952.

Dickel, L., Darmaillacq, A.S., Poirier, R., Agin, V., Bellanger, C., and Chichery, R. (2006).

Behavioural and neural maturation in the cuttlefish *Sepia officinalis*. *Vie et Milieu* 56, 89-95.

Graindorge, N., Alves, C., Darmaillacq, A.S., Chichery, R., Dickel, L. and Bellanger, C., 2006.

Effects of dorsal and ventral vertical lobe electrolytic lesions on spatial learning and locomotor activity in *Sepia officinalis*. *Behavioral neuroscience*, 120(5), p.1151.

2. Selection criteria of aged individuals. Authors report (lines 148-150) that “Aged cuttlefish that showed obvious signs of senescence were chosen for both memory experiments. These signs included a decrease in appetite, irregular flickering of chromatophore skin cells, and a decrease in reaction rate to visual stimuli.”

The Reader asks

2.1 the paradigms utilized require that the animal will make association between a visual stimulus and an ‘edible’ reward. Thus, how the choice of the test is compatible with the animal “status” and effective capability?

2.2 Authors report that animal are fed ad libitum. Authors should specify better how this is done; ad libitum means that food items are always available to the animals and this does not allow control of the amount of food taken by single individual (for example). There is no information whether before pre-training and testing phases cuttlefish have been food-deprived.

3. “we trained cuttlefish to approach a specific location in their tank” (line 152).

Size of the tank is missing, relative size when referred to number/body size of animals, home position of the animals during the test in relationship with the location of the “places”

4. It is not clear enough whether the paradigm described at (c) semantic-like memory is also referring to episodic memory. Authors should provide clear distinction in the way the tests have been designed.

The confusion also emerges when reading the Discussion (lines 304-305) “...Specifically, they were able to remember what they had eaten for breakfast, where they had eaten it, and how long ago” where the authors refer to the other one.

5. Authors do not have any indication of the status of neural circuit in the animals utilized (see also above) and in absence of details about individual performances and behavioral locomotory and other responses it is difficult to generalize the conclusions.

I suggest in any case to revise the Discussion in order to take into account the above comments.

Review form: Reviewer 2

Recommendation

Accept with minor revision (please list in comments)

Scientific importance: Is the manuscript an original and important contribution to its field?

Excellent

General interest: Is the paper of sufficient general interest?

Excellent

Quality of the paper: Is the overall quality of the paper suitable?

Excellent

Is the length of the paper justified?

Yes

Should the paper be seen by a specialist statistical reviewer?

No

Do you have any concerns about statistical analyses in this paper? If so, please specify them explicitly in your report.

No

It is a condition of publication that authors make their supporting data, code and materials available - either as supplementary material or hosted in an external repository. Please rate, if applicable, the supporting data on the following criteria.

Is it accessible?

Yes

Is it clear?

Yes

Is it adequate?

Yes

Do you have any ethical concerns with this paper?

No

Comments to the Author

The manuscript reports an interesting study focused on the potential age-related decline in measures of episodic and semantic memory in cuttlefish. The main finding is that measures of episodic and semantic memory do not decline with age in cuttlefish. The observation that episodic memory is intact at an advanced age is important for understanding the evolution of cognition and potentially opening a new model system for understanding the mechanisms that preserve this otherwise fragile memory system. I recommend that the manuscript be published

after the authors address some minor issues described below.

The authors should comment on the comparability of the semantic and episodic tasks. In some respects, the what-where-when task is more complicated as it depends on item-specific information. Yet, the semantic task used 3 times of day whereas the episodic task used 2 delay conditions. It may not be possible to fully equate the tasks, but the authors should at least comment on the issue.

L 88-91 – the authors may be interested in the study by Panoz-Brown et al 2018.

L 148-149 – were cuttlefish randomly assigned to young and old groups?

L 196 – ordinal representations of time have been well studied in other animals eg, Pizzo & Crystal 2002

References noted above:

Panoz-Brown, D., Iyer, V., Carey, L. M., Sluka, C. M., Rajic, G., Kestenman, J., Gentry, M., Brotheridge, S., Somekh, I., Corbin, H. E., Tucker, K. G., Almeida, B., Hex, S. B., Garcia, K. D., Hohmann, A. G., & Crystal, J. D. (2018). Replay of episodic memories in the rat. *Current Biology*, 28(10), 1628-1634.e1627. <https://doi.org/https://doi.org/10.1016/j.cub.2018.04.006>

Pizzo, M.J., Crystal, J.D. Representation of time in time-place learning. *Animal Learning & Behavior* 30, 387-393 (2002). <https://doi.org/10.3758/BF03195963>

Decision letter (RSPB-2021-1052.R0)

08-Jun-2021

Dear Dr Schnell:

Your manuscript has now been peer reviewed and the reviews have been assessed by an Associate Editor. The reviewers' comments (not including confidential comments to the Editor) and the comments from the Associate Editor are included at the end of this email for your reference. The reviewers and the Associate Editor and I all find quite a lot to like about your manuscript, however there were several critical issues raised that must be addressed before your manuscript can be considered further. All of these are well explained by the reviewers and nicely summarized by the AE, but to reiterate the key issues, it is necessary to include the details of the animals' husbandry and the experimental procedure, the former because we require compliance with the ARRIVE guidelines (independent of the scientists' home country guidelines) and second because right now there are not sufficient details to fully determine how the study was run. In addition, I agree with the AE that the paper will be greatly strengthened by a consideration of the cuttlefishes' natural history and some consideration of why this ability may have evolved.

We do not allow multiple rounds of revision so we urge you to make every effort to fully address all of the comments at this stage. Your manuscript will be sent back to one or more of the original reviewers for assessment. If the original reviewers are not available we may invite new reviewers. Please note that we cannot guarantee eventual acceptance of your manuscript at this stage.

Research ethics:

Use of animals and field studies:

It is a condition of publication that you make available the data and research materials supporting the results in the article. Please see our Data Sharing Policies (<https://royalsociety.org/journals/authors/author-guidelines/#data>). Datasets should be deposited in an appropriate publicly available repository and details of the associated accession number, link or DOI to the datasets must be included in the Data Accessibility section of the article (<https://royalsociety.org/journals/ethics-policies/data-sharing-mining/>). Reference(s) to datasets should also be included in the reference list of the article with DOIs (where available).

Please submit a copy of your revised paper within three weeks. If we do not hear from you within this time your manuscript will be rejected. If you are unable to meet this deadline please let us know as soon as possible, as we may be able to grant a short extension.

Best wishes,
Dr Sarah Brosnan
Editor, Proceedings B
mailto:proceedingsb@royalsociety.org

Associate Editor

Comments to Author:

Two reviewers have provided feedback on your submission and while both were in agreement that your study offers novel and interesting findings, both also ask for some revisions and clarifications.

In particular I wish to highlight the comments from Reviewer 1 who calls for greater detail in terms of your methodology and who proposes that the ARRIVE guidelines should be used as a template here (this journal promotes the use of the ARRIVE guidelines for reporting animal subjects research). I am in complete agreement with this that additional detail is required not only to understand the welfare and husbandry of the study subjects but also to facilitate reproducibility of your study protocols. For example, please describe how you trained the subjects (2b) - what training methods did you use? how long were training sessions? How many training sessions did subjects require on average and what was the variation across subjects? did you train subjects singly? Similarly, for the testing protocol, while you provide some aspects of the methods (e.g., how far apart the feeding stations were) other elements (such as tank size, stimuli size etc.) were lacking. I think some more detailed schematics in addition to the current Fig 1 might help.

Finally, and while you touch on this in your introduction, I think your Discussion would benefit from a more detailed consideration of the evolutionary processes that might have shaped the results you find i.e. what is it about the cuttlefishes natural history and native ecology or sociality that might help interpret these results. This broader perspective - beyond narrowly focussing on the cognitive underpinnings - would be welcome given the broad scope of this journal.

Reviewer(s)' Comments to Author:

Referee: 1

Comments to the Author(s)

I read the manuscript "Episodic-like memory is preserved with age in cuttlefish" and found it an interesting study exploring episodic memory in the cephalopod mollusc *Sepia officinalis*.

First of all, I have to say that despite the study does not require ethical approval because experiments on live animals have been carried out in the USA, there are important details about care, maintenance and use of live animals for research purposes that should be reported. This is mandatory according to ARRIVE Guidelines - that this Journal and the Royal Society that states "Authors are expected to comply with the 'Animal Research: Reporting In Vivo Experiments' (ARRIVE) guidelines. These have been developed by NC3Rs to improve standards of reporting to ensure that the data from animal experiments can be fully scrutinised and utilised. Relevant

information should be included in the appropriate section of the article, as outlined in the ARRIVE guidelines" (see <https://royalsociety.org/journals/ethics-policies/research-ethics/>).

I strongly recommend the Authors to revise their manuscript attentively, in order to report all the necessary details about this study as required by ARRIVE Checklist and the best practice of in vivo studies reporting (see also PREPARE: Smith, A.J., Clutton, R.E., Lilley, E., Hansen, K.E.A., and Brattelid, T. (2018). PREPARE: guidelines for planning animal research and testing. *Laboratory Animals* 52(2), 135-141. doi: 10.1177/0023677217724823.). I recommend to provide details (to mention some) about: i. the individual/group housing of the animals between experimental trials; ii. body size of the animals (in addition to age group); iii. size and shape and all required details about the holding and experimental tanks; iv. monitoring of seawater quality, v. number of subjects excluded (if any).

I would like also to remind that following the "Guidelines for the treatment of animals in behavioural research and teaching" studies carried out using captive animals "may mean obtaining them from the wild and necessarily involve confinement" require attention in assuring best practice and compliance with the principles of keeping animals in an optimal welfare status, including manipulation/handling.

This reviewer is fully aware of the long, well established, consolidated practice in housing live cephalopods at the MBL, but I reiterate that the overall quality of the scientific work using these species will enormous improve if transparency about animal care and maintenance and details about experimental settings will be provided, despite not required by legislation in non-EU countries.

The study is an interesting one exploring time-age capacity for episodic and semantic memories in the cuttlefish. As reported by Authors, known age-related decline in episodic memory "is thought to stem from deteriorating function of the hippocampus in the brain. Whether episodic memory can deteriorate with age in species that lack a hippocampus". Testing this research question in cuttlefish offers interesting advantages.

Authors tested semantic- and episodic-like memories in sub-adults and aged cuttlefish. The semantic-like memory task was designed to let the animal to learn that the location of a food resource in relation with the time of the day. In the episodic-like memory task, the paradigm designed required the solution of a foraging task (what-where-when information about a past event with unique spatiotemporal features). Performance of the semantic-like memory task resulted comparable across age groups. However, with the episodic memory Authors found that aged-adults reached the success criterion faster than sub-adults. Authors conclude that contrary to what known in other species, episodic-like memory is preserved in aged cuttlefish, "suggesting that memory deterioration is delayed in this species".

Independently from what anticipated in the previous paragraphs, I am in the unfortunate situation of not being able to recommend the acceptance of the manuscript in its current form. I am going to provide the following comments for Authors to consider in their substantial revision.

1. The 'statement' (line 28) "Cuttlefish are molluscs that lack a hippocampus" is not accurate. Despite the classic knowledge by Young and colleagues, that I recommend to refer to, I suggest Authors to reconsider their statement and refer to:

Shomrat, T., Graindorge, N., Bellanger, C., Fiorito, G., Loewenstein, Y., and Hochner, B. (2011). Alternative Sites of Synaptic Plasticity in Two Homologous "Fan-out Fan-in" Learning and Memory Networks. *Current Biology* 21, 1773-1782.

Shigeno, S., Andrews, P.L.R., Ponte, G., and Fiorito, G. (2018). Cephalopod Brains: An Overview of Current Knowledge to Facilitate Comparison With Vertebrates. *Front. Physiol.* 9, 952. doi: 10.3389/fphys.2018.00952.

Dickel, L., Darmaillacq, A.S., Poirier, R., Agin, V., Bellanger, C., and Chichery, R. (2006). Behavioural and neural maturation in the cuttlefish *Sepia officinalis*. *Vie et Milieu* 56, 89-95.

Graindorge, N., Alves, C., Darmaillacq, A.S., Chichery, R., Dickel, L. and Bellanger, C., 2006. Effects of dorsal and ventral vertical lobe electrolytic lesions on spatial learning and locomotor activity in *Sepia officinalis*. *Behavioral neuroscience*, 120(5), p.1151.

2. Selection criteria of aged individuals. Authors report (lines 148-150) that “Aged cuttlefish that showed obvious signs of senescence were chosen for both memory experiments. These signs included a decrease in appetite, irregular flickering of chromatophore skin cells, and a decrease in reaction rate to visual stimuli.”

The Reader asks

2.1 the paradigms utilized require that the animal will make association between a visual stimulus and an ‘edible’ reward. Thus, how the choice of the test is compatible with the animal “status” and effective capability?

2.2 Authors report that animal are fed ad libitum. Authors should specify better how this is done; ad libitum means that food items are always available to the animals and this does not allow control of the amount of food taken by single individual (for example). There is no information whether before pre-training and testing phases cuttlefish have been food-deprived.

3. “we trained cuttlefish to approach a specific location in their tank” (line 152).

Size of the tank is missing, relative size when referred to number/body size of animals, home position of the animals during the test in relationship with the location of the “places”

4. It is not clear enough whether the paradigm described at (c) semantic-like memory is also referring to episodic memory. Authors should provide clear distinction in the way the tests have been designed.

The confusion also emerges when reading the Discussion (lines 304-305) “...Specifically, they were able to remember what they had eaten for breakfast, where they had eaten it, and how long ago” where the authors refer to the other one.

5. Authors do not have any indication of the status of neural circuit in the animals utilized (see also above) and in absence of details about individual performances and behavioral locomotory and other responses it is difficult to generalize the conclusions.

I suggest in any case to revise the Discussion in order to take into account the above comments.

Referee: 2

Comments to the Author(s)

The manuscript reports an interesting study focused on the potential age-related decline in measures of episodic and semantic memory in cuttlefish. The main finding is that measures of episodic and semantic memory do not decline with age in cuttlefish. The observation that episodic memory is intact at an advanced age is important for understanding the evolution of cognition and potentially opening a new model system for understanding the mechanisms that preserve this otherwise fragile memory system. I recommend that the manuscript be published after the authors address some minor issues described below.

The authors should comment on the comparability of the semantic and episodic tasks. In some respects, the what-where-when task is more complicated as it depends on item-specific information. Yet, the semantic task used 3 times of day whereas the episodic task used 2 delay conditions. It may not be possible to fully equate the tasks, but the authors should at least comment on the issue.

L 88-91 – the authors may be interested in the study by Panoz-Brown et al 2018.

L 148-149 – were cuttlefish randomly assigned to young and old groups?

L 196 – ordinal representations of time have been well studied in other animals eg, Pizzo & Crystal 2002

References noted above:

Panoz-Brown, D., Iyer, V., Carey, L. M., Sluka, C. M., Rajic, G., Kestenman, J., Gentry, M., Brotheridge, S., Somekh, I., Corbin, H. E., Tucker, K. G., Almeida, B., Hex, S. B., Garcia, K. D., Hohmann, A. G., & Crystal, J. D. (2018). Replay of episodic memories in the rat. *Current Biology*, 28(10), 1628-1634.e1627. <https://doi.org/https://doi.org/10.1016/j.cub.2018.04.006>

Pizzo, M.J., Crystal, J.D. Representation of time in time-place learning. *Animal Learning & Behavior* 30, 387–393 (2002). <https://doi.org/10.3758/BF03195963>

Author's Response to Decision Letter for (RSPB-2021-1052.R0)

See Appendix A.

Decision letter (RSPB-2021-1052.R1)

12-Jul-2021

Dear Dr Schnell

I am pleased to inform you that your manuscript RSPB-2021-1052.R1 entitled "Episodic-like memory is preserved with age in cuttlefish" has been accepted for publication in *Proceedings B* pending some minor revisions, listed below. I invite you to respond to the referee(s)' comments and revise your manuscript. Because the schedule for publication is very tight, it is a condition of publication that you submit the revised version of your manuscript within 7 days. If you do not think you will be able to meet this date please let us know.

1) A text file of the manuscript (doc, txt, rtf or tex), including the references, tables (including captions) and figure captions. Please remove any tracked changes from the text before submission. PDF files are not an accepted format for the "Main Document".

2) A separate electronic file of each figure (tiff, EPS or print-quality PDF preferred). The format should be produced directly from original creation package, or original software format. PowerPoint files are not accepted.

3) Electronic supplementary material: this should be contained in a separate file and where possible, all ESM should be combined into a single file. All supplementary materials accompanying an accepted article will be treated as in their final form. They will be published alongside the paper on the journal website and posted on the online figshare repository. Files on figshare will be made available approximately one week before the accompanying article so that the supplementary material can be attributed a unique DOI.

Sincerely,

Dr Sarah Brosnan

Associate Editor:

Board Member

Comments to Author:

Thank you very much for so carefully responding to the reviewers' feedback. I find the methods much more clear and I appreciate the expanded ethics section.

I only have three minor outstanding comments:

1. Although the reader can parse it from the manuscript, I think it would be helpful to state explicitly in the raw data what the values in the cells within each tab refer to. While this has been done in one tab (proportion of correct responses), this would be valuable for all tabs.
2. As the data were collected across two different time points (2016 and 2018), please state explicitly in your methods (or supplemental materials) whether there were any changes in lab protocols across the years that may affect the results. I am assuming not, but it would be worth stating this (e.g., changes in personnel, lighting schedules, food etc.)
3. As you only tested a relatively small sample size ($N=6$ per cohort) please state/acknowledge this in your abstract and Discussion.

Author's Response to Decision Letter for (RSPB-2021-1052.R1)

See Appendix B.

Decision letter (RSPB-2021-1052.R2)

22-Jul-2021

Dear Dr Schnell

I am pleased to inform you that your manuscript entitled "Episodic-like memory is preserved with age in cuttlefish" has been accepted for publication in Proceedings B.

Data Accessibility section

Open Access

Paper charges

Sincerely,

Proceedings B

Associate Editor:

Comments to Author:

Thank you very much for making all of the suggested edits and clarifications that I proposed.

Appendix A

Manuscript ID RSPB-2021-1052

Episodic-like memory is preserved with age in cuttlefish

Thank you for inviting us to revise this manuscript. We have conducted revisions and believe that our manuscript has improved thanks to the constructive feedback provided by the editor and the reviewers – for this we are grateful and thank them for their time and effort. For your reference, we have provided a response below each of the reviewers' comments.

Associate Editor

Comments to Author:

Two reviewers have provided feedback on your submission and while both were in agreement that your study offers novel and interesting findings, both also ask for some revisions and clarifications.

In particular I wish to highlight the comments from Reviewer 1 who calls for greater detail in terms of your methodology and who proposes that the ARRIVE guidelines should be used as a template here (this journal promotes the use of the ARRIVE guidelines for reporting animal subjects research). I am in complete agreement with this that additional detail is required not only to understand the welfare and husbandry of the study subjects but also to facilitate reproducibility of your study protocols. For example, please describe how you trained the subjects (2b) - what training methods did you use? how long were training sessions? How many training sessions did subjects require on average and what was the variation across subjects? did you train subjects singly? Similarly, for the testing protocol, while you provide some aspects of the methods (e.g., how far apart the feeding stations were) other elements (such as tank size, stimuli size etc.) were lacking. I think some more detailed schematics in addition to the current Fig 1 might help.

Authors' response: Thanks for these insightful comments. To show that our practices coincide with ARRIVE guidelines we have added extra details in our methods section as electronic supplementary material. We have also added more details about our training processes to facilitate reproducibility of our study protocols, these can be found in the electronic supplementary materials. Finally, we have added an extra schematic figure to help readers visualise our methods in the semantic-like memory task (see Figure 1).

Finally, and while you touch on this in your introduction, I think your Discussion would benefit from a more detailed consideration of the evolutionary processes that might have shaped the results you find i.e., what is it about the cuttlefishes natural history and native ecology or sociality that might help interpret these results. This broader perspective - beyond narrowly focussing on the cognitive underpinnings - would be welcome given the broad scope of this journal.

Authors' response: We have devoted a paragraph on this subject previously in the discussion section (see lines: 373–394) and we believe it is sufficient coverage for this issue lest it become too speculative. Specifically, we argue that resistance to age-related decline to preserve complex learning and memory might have been positively selected for because it enables individuals to recall specific spatiotemporal

features of past mating events, especially given the relatively short lifespan of these animals.

Referee: 1

Comments to the Author(s)

I read the manuscript “Episodic-like memory is preserved with age in cuttlefish” and found it an interesting study exploring episodic memory in the cephalopod mollusc *Sepia officinalis*.

First of all, I have to say that despite the study does not require ethical approval because experiments on live animals have been carried out in the USA, there are important details about care, maintenance and use of live animals for research purposes that should be reported. This is mandatory according to ARRIVE Guidelines – that this Journal and the Royal Society that states “Authors are expected to comply with the 'Animal Research: Reporting In Vivo Experiments' (ARRIVE) guidelines. These have been developed by NC3Rs to improve standards of reporting to ensure that the data from animal experiments can be fully scrutinised and utilised. Relevant information should be included in the appropriate section of the article, as outlined in the ARRIVE guidelines” (see <https://royalsociety.org/journals/ethics-policies/research-ethics/>).

I strongly recommend the Authors to revise their manuscript attentively, in order to report all the necessary details about this study as required by ARRIVE Checklist and the best practice of in vivo studies reporting (see also PREPARE: Smith, A.J., Clutton, R.E., Lilley, E., Hansen, K.E.A., and Brattelid, T. (2018). PREPARE: guidelines for planning animal research and testing. *Laboratory Animals* 52(2), 135-141. doi: 10.1177/0023677217724823.). I recommend to provide details (to mention some) about: i. the individual/group housing of the animals between experimental trials; ii. body size of the animals (in addition to age group); iii. size and shape and all required details about the holding and experimental tanks; iv. monitoring of seawater quality, v. number of subjects excluded (if any).

I would like also to remind that following the “Guidelines for the treatment of animals in behavioural research and teaching” studies carried out using captive animals “may mean obtaining them from the wild and necessarily involve confinement” require attention in assuring best practice and compliance with the principles of keeping animals in an optimal welfare status, including manipulation/handling.

This reviewer is fully aware of the long, well established, consolidated practice in housing live cephalopods at the MBL, but I reiterate that the overall quality of the scientific work using these species will enormous improve if transparency about animal care and maintenance and details about experimental settings will be provided, despite not required by legislation in non-EU countries.

*Authors’ response: We thank the reviewer for these suggestions and we have provided many more details on this subject. The fact that *Sepia officinalis* has been*

cultured annually from feral eggs from England for 20 consecutive years at the MBL with very high survival rates for the entire life cycle is strong evidence for the quality of animal care delivered to this animal model. The tank designs and water flow systems are tailor made to this species and are based largely on our field studies on this species. We only provide one recent reference for the culture methods (Panetta et al. 2017 - #58 in references) and we expect interested parties to follow the literature cited in the paper to obtain previous culture papers on S. officinalis).

I am going to provide the following comments for Authors to consider in their substantial revision.

1. The ‘statement’ (line 28) “Cuttlefish are molluscs that lack a hippocampus” is not accurate. Despite the classic knowledge by Young and colleagues, that I recommend to refer to, I suggest Authors to reconsider their statement and refer to:

Shomrat, T., Graindorge, N., Bellanger, C., Fiorito, G., Loewenstein, Y., and Hochner, B. (2011). Alternative Sites of Synaptic Plasticity in Two Homologous "Fan-out Fan-in" Learning and Memory Networks. *Current Biology* 21, 1773-1782.

Shigeno, S., Andrews, P.L.R., Ponte, G., and Fiorito, G. (2018). Cephalopod Brains: An Overview of Current Knowledge to Facilitate Comparison With Vertebrates. *Front. Physiol.* 9, 952. doi: 10.3389/fphys.2018.00952.

Dickel, L., Darmaillacq, A.S., Poirier, R., Agin, V., Bellanger, C., and Chichery, R. (2006). Behavioural and neural maturation in the cuttlefish *Sepia officinalis*. *Vie et Milieu* 56, 89-95.

Graindorge, N., Alves, C., Darmaillacq, A.S., Chichery, R., Dickel, L. and Bellanger, C., 2006. Effects of dorsal and ventral vertical lobe electrolytic lesions on spatial learning and locomotor activity in *Sepia officinalis*. *Behavioral neuroscience*, 120(5), p.1151.

Authors’ response: We agree with Reviewer 1 that the vertical lobe in the cuttlefish brain presents similarities in connectivity and functionality to the hippocampus formation in the vertebrate brain. However, it is not referred to as a hippocampus in any of the peer-reviewed literature including the literature listed above and thus put simply, the cuttlefish does lack a hippocampus. Consequently, we respectively have decided to keep this statement. As we are limited by word restrictions in the abstract, we are unable to add the extra detail about the similarities between the vertical lobe and the hippocampal formation in the abstract. Nevertheless, we draw attention to the similarities between the hippocampus and the vertical lobe in the introduction and cite the literature that Reviewer 1 has listed here (see lines: 117-120).

2. Selection criteria of aged individuals. Authors report (lines 148-150) that “Aged cuttlefish that showed obvious signs of senescence were chosen for both memory experiments. These signs included a decrease in appetite, irregular flickering of chromatophore skin cells, and a decrease in reaction rate to visual stimuli.”

The Reader asks

2.1 the paradigms utilized require that the animal will make association between a visual stimulus and an ‘edible’ reward. Thus, how the choice of the test is compatible with the animal “status” and effective capability?

Authors' response: While the reaction rate was slower towards visual stimuli in aged-adults, the individuals were still able to recognise visual stimuli, swim towards them and eat three times a day. Aged-adults that ignored visual stimuli and were completely uninterested in food or ate less than three times a day were deemed unfit for this experiment. These details are now included in the electronic supplementary materials.

2.2 Authors report that animal are fed ad libitum. Authors should specify better how this is done; ad libitum means that food items are always available to the animals and this does not allow control of the amount of food taken by single individual (for example).

Authors' response: We apologise for not having clearer timeframes for feeding regimes. We have now included approximate feeding times outside of experimental periods in the methods section (see line 54)

There is no information whether before pre-training and testing phases cuttlefish have been food-deprived.

Authors' response: To ensure the cuttlefish were motivated to participate in the experiments, the 0900 h and 1200 h feeding sessions were omitted. The amount of food they obtained during training and testing replaced the food acquired during these feeding times. If a subject did not participate in the trials the subject was offered extra food during the 1630 h feed. We have now included these details in the methods section (see lines: 156-160).

3. “we trained cuttlefish to approach a specific location in their tank” (line 152). Size of the tank is missing, relative size when referred to number/body size of animals, home position of the animals during the test in relationship with the location of the “places”

Authors' response: We have now included the size of the tank and the size of the animals in the electronic supplementary materials. We have also included details about the starting position of the subject with respect to the semantic-like memory task (see lines: 183-185 as well as a new figure (figure 1)).

4. It is not clear enough whether the paradigm described at (c) semantic-like memory is also referring to episodic memory. Authors should provide clear distinction in the way the tests have been designed.

The confusion also emerges when reading the Discussion (lines 304-305) “...Specifically, they were able to remember what they had eaten for breakfast, where they had eaten it, and how long ago” where the authors refer to the other one.

*Authors' response: Apologies for the confusion. The difference lies in the uniqueness of the memory. In the semantic-like task they are recalling learnt information that is not unique to a specific time or place but is fixed across both training and testing days. By contrast, in the episodic-like memory task they are recalling a specific memory that is unique to each test day. We have now included this in the introduction section (see lines: 133-137). We have also changed the wording in the discussion where we elaborate on the episodic-like memory task. This sentence now reads: ‘Specifically, they were able to remember what they had eaten during a **unique** breakfast event, where....’ (see line 313)*

5. Authors do not have any indication of the status of neural circuit in the animals utilized (see also above) and in absence of details about individual performances and behavioral locomotory and other responses it is difficult to generalize the conclusions.

I suggest in any case to revise the Discussion in order to take into account the above comments.

Authors' response: We agree with Reviewer 1, we do not know the neural status of the animals. We have added a couple of sentences and further detail (see lines 366–370) in the discussion to demonstrate that we are not attempting to draw concrete conclusions using our behavioural results. Rather, we are suggesting that cognitive aging in cuttlefish might not follow a gradual time course as observed in other studied species.

Referee: 2

Comments to the Author(s)

The manuscript reports an interesting study focused on the potential age-related decline in measures of episodic and semantic memory in cuttlefish. The main finding is that measures of episodic and semantic memory do not decline with age in cuttlefish. The observation that episodic memory is intact at an advanced age is important for understanding the evolution of cognition and potentially opening a new model system for understanding the mechanisms that preserve this otherwise fragile memory system. I recommend that the manuscript be published after the authors address some minor issues described below.

The authors should comment on the comparability of the semantic and episodic tasks. In some respects, the what-where-when task is more complicated as it depends on item-specific information. Yet, the semantic task used 3 times of day whereas the episodic task used 2 delay conditions. It may not be possible to fully equate the tasks, but the authors should at least comment on the issue.

Authors' response: Thanks for this comment. We now mention this in the discussion (see lines: 343–351). 'Notice that the tasks include a different number of delay conditions. Specifically, the semantic-like memory task involves 3 delay conditions whereas the episodic-like memory task involves only 2 delay conditions. The difference in the number of delay conditions across both tasks means that we were unable to equate the complexity of the tasks, making it difficult to draw comparisons of performance. Future research could increase comparability by designing tasks that involve and equal number of delay conditions.

L 88-91 – the authors may be interested in the study by Panoz-Brown et al 2018.

Authors' response: Thank you for bringing our attention to this interesting study, we have now including these new findings in the introduction (see lines: 91-92).

L 148-149 – were cuttlefish randomly assigned to young and old groups?

Authors' response: Yes, cuttlefish were pseudo-randomly assigned to young and old groups in both experiments. We have now added this detail in the electronic supplementary materials.

L 196 – ordinal representations of time have been well studied in other animals eg, Pizzo & Crystal 2002

Author's response: We have now included this reference in our manuscript (see lines: 209; 214).

References noted above:

Panoz-Brown, D., Iyer, V., Carey, L. M., Sluka, C. M., Rajic, G., Kestenman, J., Gentry, M., Brotheridge, S., Somekh, I., Corbin, H. E., Tucker, K. G., Almeida, B., Hex, S. B., Garcia, K. D., Hohmann, A. G., & Crystal, J. D. (2018). Replay of episodic memories in the rat. *Current Biology*, 28(10), 1628-1634.e1627. <https://doi.org/https://doi.org/10.1016/j.cub.2018.04.006>

Pizzo, M.J., Crystal, J.D. Representation of time in time-place learning. *Animal Learning & Behavior* 30, 387–393 (2002). <https://doi.org/10.3758/BF03195963>

Appendix B

Manuscript ID RSPB-2021-1052

Episodic-like memory is preserved with age in cuttlefish

Thank you for accepting our manuscript. We have conducted the three minor revisions that you have requested and provided answers to your queries below. We look forward to being published in Proceedings of the Royal Society B.

Associate Editor

Comments to Author:

I only have three minor outstanding comments:

1. Although the reader can parse it from the manuscript, I think it would be helpful to state explicitly in the raw data what the values in the cells within each tab refer to. While this has been done in one tab (proportion of correct responses), this would be valuable for all tabs.

Authors' response: We agree and have now amended the raw data in the ESM to include explicit labels for our data.

2. As the data were collected across two different time points (2016 and 2018), please state explicitly in your methods (or supplemental materials) whether there were any changes in lab protocols across the years that may affect the results. I am assuming not, but it would be worth stating this (e.g., changes in personnel, lighting schedules, food etc.)

Authors' response: We have now included these details in the supplementary material at the end of the fourth paragraph, which follows our methods that outline that cuttlefish cohorts were tested across different years for the episodic-like memory experiment.

3. As you only tested a relatively small sample size (N=6 per cohort) please state/acknowledge this in your abstract and Discussion.

Authors' response: We have now included these details in both the abstract and discussion (see lines: 29; 351–353).